

# Formation and evolution of newly formed glaciovolcanic caves in the crater of Mount St. Helens, Washington, USA

Linda Sobolewski[1], Christian Stenner[2], Charlotte Hüser[1], Tobias Berghaus[1], Eddy Cartaya[3], Andreas Pflitsch[1]

[1]Institute of Geography, Ruhr-University Bochum, Bochum, 44801, Germany
[2]Alberta Speleological Society, Calgary, AB T2Z2E3, Canada
[3]Glacier Cave Explorers, Redmond, OR 97756, USA

*Correspondence to*: Linda Sobolewski (linda.sobolewski@ruhr-uni-bochum.de)

**Abstract.** A new and extensive system of glaciovolcanic caves has developed around the 2004-2008 lava dome in the crater
of Mount St. Helens, Washington, USA. These systems offer a rare view into a subglacial environment and lead to a better understanding of how glaciers and active volcanoes interact. Here, we present first results from geodetic and optical surveys done between 2014 and 2019 as well as climatologic studies performed between 2017 and 2019. Our data show that volcanic activity has altered subglacial morphology in numerous ways and formed new cave systems that are strongly affected by heat flux from several subglacial fumaroles. More than 2.3 km of cave passages now form a circumferential pattern around the
dome, some several hundred meters long. Air and fumarole temperature measurements were conducted in two specific caves. Whereas air temperatures reveal a strong seasonal dependency, fumarole temperatures are affected to a minor extent and are primarily regulated by changes in volcanic heat flux or the contribution of glacial melt. Related studies from Mount Hood, Oregon, and Mount Rainier, Washington, are used as comparison between glaciovolcanic cave systems. Fumarolic heat and resulting microclimates enable further genesis of this dynamic system. Already one of the largest worldwide, it is very likely
that the system will continue to expand. As Mount St. Helens is the Cascade Volcano most likely to erupt again in the near future, these caves represent a unique laboratory to understand glaciovolcanic interactions, monitor indicators of recurring volcanic activity and to predict related hazards.

## 1 Introduction

Most volcanoes of the Cascade Volcanic Arc (Fig. 1) are characterized by sizeable glaciers, the most voluminous glaciers
existing in Washington State (Harris, 2005). It is known that some of these glaciers contain large cave passages. Mount Rainier, hosting two ice-filled summit craters (Kiver and Mumma, 1971), contains the world's largest glacier cave system on an active volcanic edifice (Florea et al., 2020, in review; Zimbelman et al., 2000). Other examples include Mount Hood's Sandy Glacier (Pflitsch et al., 2017), and the glacier in the crater of Mount St. Helens (Anderson et al., 1998; Anderson and Vining, 1999), since 2006 officially called Crater Glacier (Walder et al., 2008). Apart from the Cascade Volcanoes, numerous caves also
appear on Mount Erebus, Antarctica (Curtis and Kyle, 2011). These glaciovolcanic caves, often referred to as glacier caves, usually form as a result of several mechanisms such as climatic impacts, glacier movement, geothermal activity, or a





combination thereof (Badino et al., 2007; Benn and Evans, 2010). Glaciovolcanic caves, their importance to understand the volcano's hydrothermal and magmatic system, or their significance to serve as important analogues for extraterrestrial phenomena (Curtis and Kyle, 2011; Zimbelman et al., 2000) have been identified and discussed in literature. First studies

comprise research on the summit craters of Mount Rainier (Kiver and Mumma, 1971), and investigations of the geothermal ice caves on Mount Erebus (Giggenbach, 1976). However, research in these harsh environments hold numerous hazards like volcanic gases, the potential beginning of an eruption, or any instability of the volcanic edifice. A problem unique to the Crater Glacier is the abundance of loose rock with an average content of 15 %, deriving from the surrounding amphitheater walls (Walder et al., 2008). Schilling et al., (2004) even suggested one-third of the glacier's volume to be rock debris from abounding

rock avalanches.

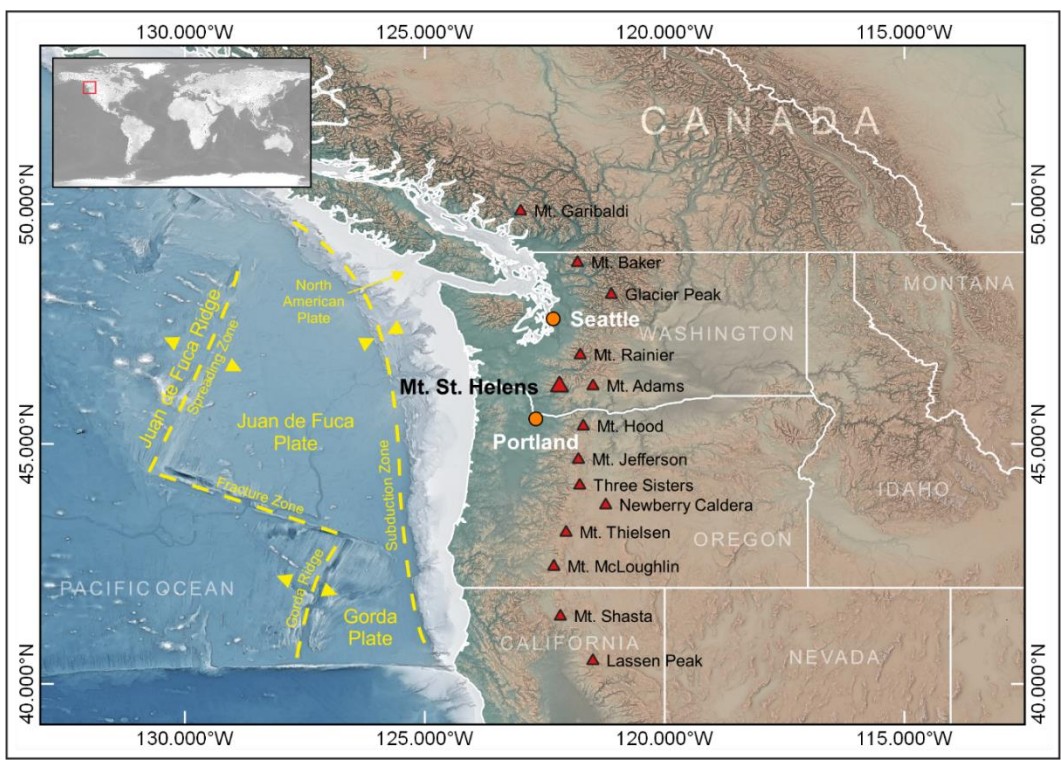

**Figure 1:** The Cascade Volcanic Arc with main geological structures and major cities. Basic map: General Bathymetric Chart of the Oceans (GEBCO). Inset: Natural Earth.


On Mount St. Helens the first descriptions of glaciovolcanic caves in firn ice were given by Anderson et al., (1998). These firn caves extended along the southwest to the southeast side of the 1980-1986 lava dome. As the reawakening of the volcano between 2004 and 2008 led to the growth of a new dome south of the previous one, the former cave systems were disrupted and obliterated. Evidence of new glaciovolcanic caves within the crater came from aerial observations in 2012, revealing that

a large chasm had formed in the glacier between the crater walls and the new dome. Our project began when this opening was





investigated and descended by our team for the first time in 2014. Continued investigations in the crater around the 2004-2008 lava dome from 2014 to 2019 have located 10 distinct glaciovolcanic caves (Fig. 2). Repeated geodetic and optical surveys allow comparisons of passage extent and volume to be made, providing a unique opportunity to observe their formation and evolution over time.


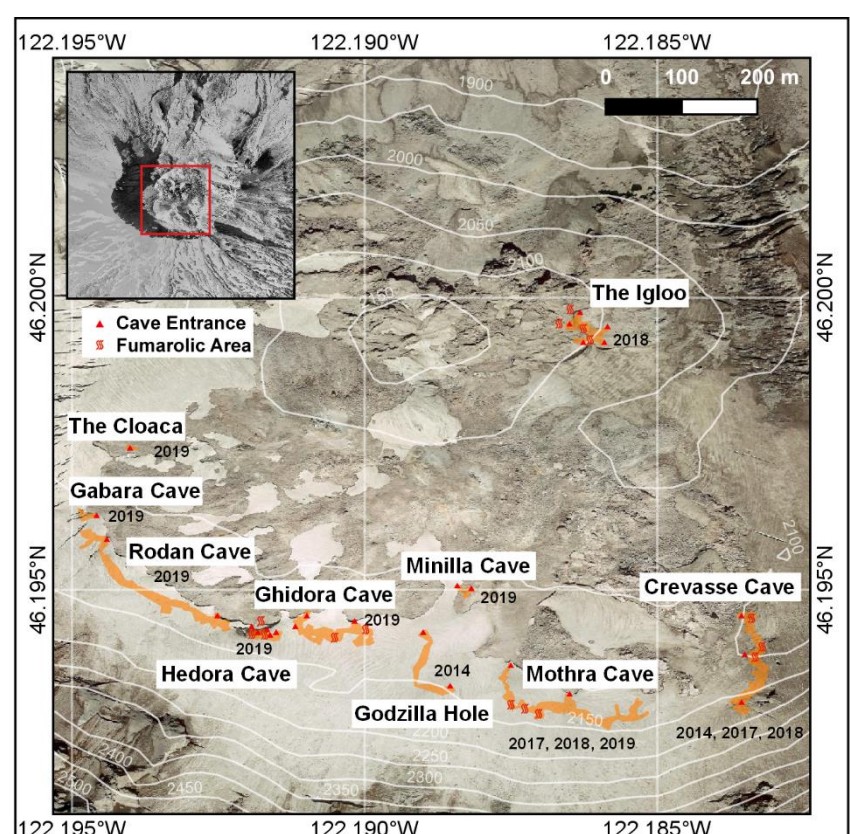

**Figure 2:** Location of glaciovolcanic cave systems around the 2004-2008 lava dome in the crater of Mount St. Helens with main entrances. Black numbers indicate the time of survey. Spots of concentrated fumarolic activity are indicated when clearly identified for individual cave systems. Image taken in 2018, © Google Earth. The inset illustrates the dimension of the crater. Image: High Resolution Orthoimagery
(2006) from USGS Earth Explorer.

We present the first overview of glaciovolcanic cave systems circumnavigating the 2004-2008 lava dome. We discuss the formation and evolution of cave systems recorded over six years through tacheometric survey methods and discuss the results of air and fumarole temperature measurements from 2017 to 2019 in two distinct caves. We also present essential cave

characteristics identified by direct observations. The caves experience seasonal changes and although they share this feature with other cave systems, e.g. on Mount Rainier (Florea et al., 2020, in review), several features unique to the cave systems on Mount St. Helens have been discovered. Our data reveal that these caves show incredible dynamic growth compared to other





glaciovolcanic cave systems and are trending to continue expansion. This paper provides a general summary to introduce main areas of research and only represents selected parts of our work.

**2 Geological setting**

Mount St. Helens, an active andesite-dacite volcano (Anderson and Vining, 1999), is located in Washington State in the Cascade Range, a magmatic arc resulting from the subduction of the oceanic Juan de Fuca Plate beneath the continental North American Plate (Miller and Cowan, 2017). Coming into existence between 40,000 and 50,000 years ago, Mount St. Helens remains the youngest of five major stratovolcanoes which developed in Washington State (Harris, 2005). In 1980 activity at

the volcano was rekindled in terms of a catastrophic explosive eruption which dramatically modified the surrounding landscape and the volcano itself (Voight et al., 1983). The eruption caused a loss of approximately 70 % of the volcano's ice cover (Brugman and Post, 1981) and resulted in a 1.6 x 3.5 km horseshoe-shaped and north-facing crater (Anderson et al., 1998). Despite a relatively low elevation of the crater floor, this area represents an important accumulation zone for snow, rime, and avalanche deposits throughout the year due to deep shade and insulating characteristics of dust layers (Schilling et al., 2004).

From 1980 until 1986, the unrest of Mount St. Helens has generated a new lava dome (Fig. 3), episodically growing on the new crater floor (Swanson and Holcomb, 1990). Simultaneously, a progressive accumulation of snow heralded the beginning of a new glacier to form. This process was accompanied by a series of 16 dome-building eruptions (Anderson et al., 1998) and at least six smaller ash-producing explosions with minor fallout between 1989-1991 (Vallance et al., 2010). Another unrest started in September 2004, when a second lava-dome-building eruption initiated in the crater. This renewed activity greatly

influenced the morphology of the glacier, which had already grown to a thickness of 150 m. Ongoing activity through 2008 bisected, fractured, and compressed the glacier, doubling the surviving east and west arms in thickness and increasing the flow rate (Scott et al., 2008; Vallance et al., 2010). The formation of the new lava dome, the whaleback spine, also involved small explosions as well as plumes of volcanic ash and gases (Iverson et al., 2006). Even though the volcano is quiescent at the moment, it is supposed that it will erupt again this century (Vallance et al., 2010). During the past 4,000 years it has been the

most active volcano in the Cascade Volcanic Arc (Crandell and Mullineaux, 1978), and according to the Cascade Volcano Observatory currently considered as the Cascade Volcano most likely to erupt again in the near future.



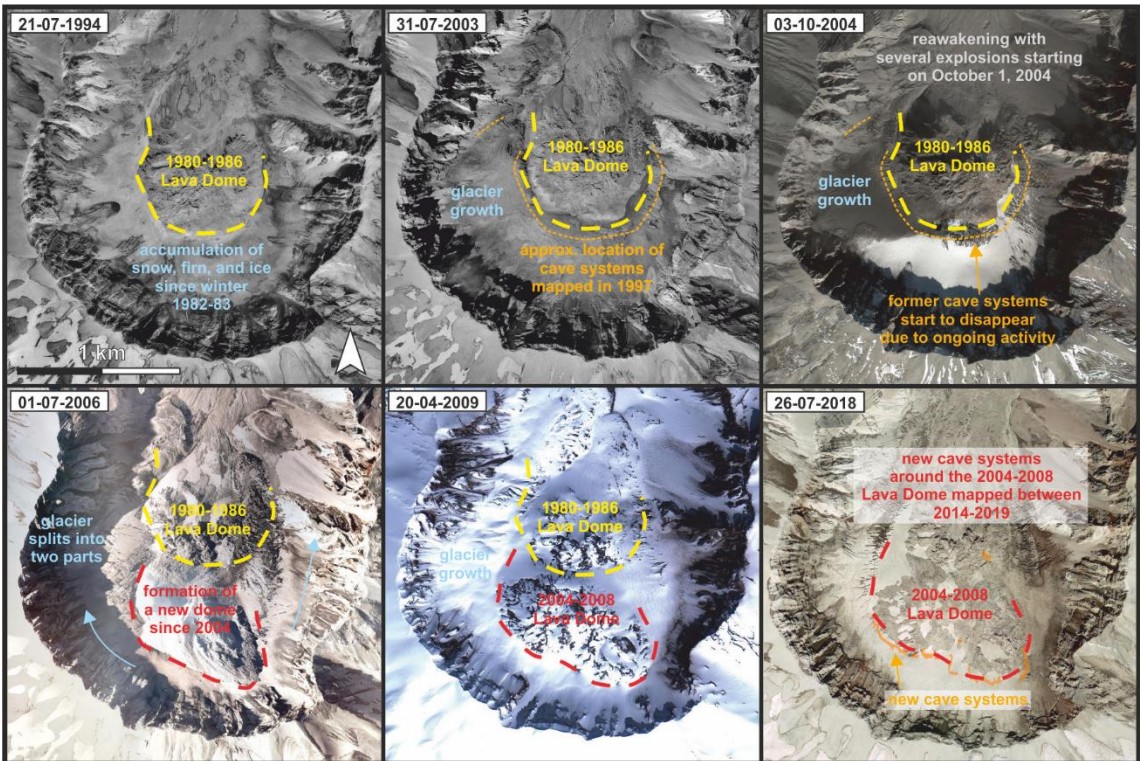

**Figure 3:** Development of crater and Crater Glacier indicating main morphological changes from 1994 to 2018 due to the generation of a new lava dome and glacier growth. Images: Google Earth, with date indicated on each image. © Google Earth.

## 3 Methods

### 3.1 Cave survey

Travel on the Crater Glacier and cave exploration requires mountaineering on glacier ice, ice climbing techniques, and the capability to ascend and descend cave entrances and passages. Crevasse hazards ranged from minimal to extensive on the east and west side the 2004-2008 lava dome, respectively. To mitigate some of the risks of severe weather, rockslides, and firn collapse, expeditions were confined to May and June. Potential cave entrances were located by reconnaissance via satellite imagery and by ground observations. Exploration to locate caves was not undertaken in the area forward of the 1980-1986 lava dome as there the glacier is heavily fractured.

### 3.1.1 Data collection in the crater

Tacheometric surveys were used to record the extent of 10 discovered glaciovolcanic caves (Fig. 2). Furthermore, surveys were used to produce fixed, recoverable stations and record the location of fumarole and temperature loggers inside Mothra Cave (Fig. 4) and Crevasse Cave (Fig. 5). GPS georeferenced stations were recorded at each cave entrance. Cave survey data





were collected using DistoX and DistoX2 to generate distance, azimuth, and inclination and to compute passage volumes at

each station. The device is a Leica Laser Disto customized with a 3-axis electronic compass, clinometer, and a wireless

Bluetooth connection (Heeb, 2009) to communicate with PDA or PC based applications to manage and visualize the data.

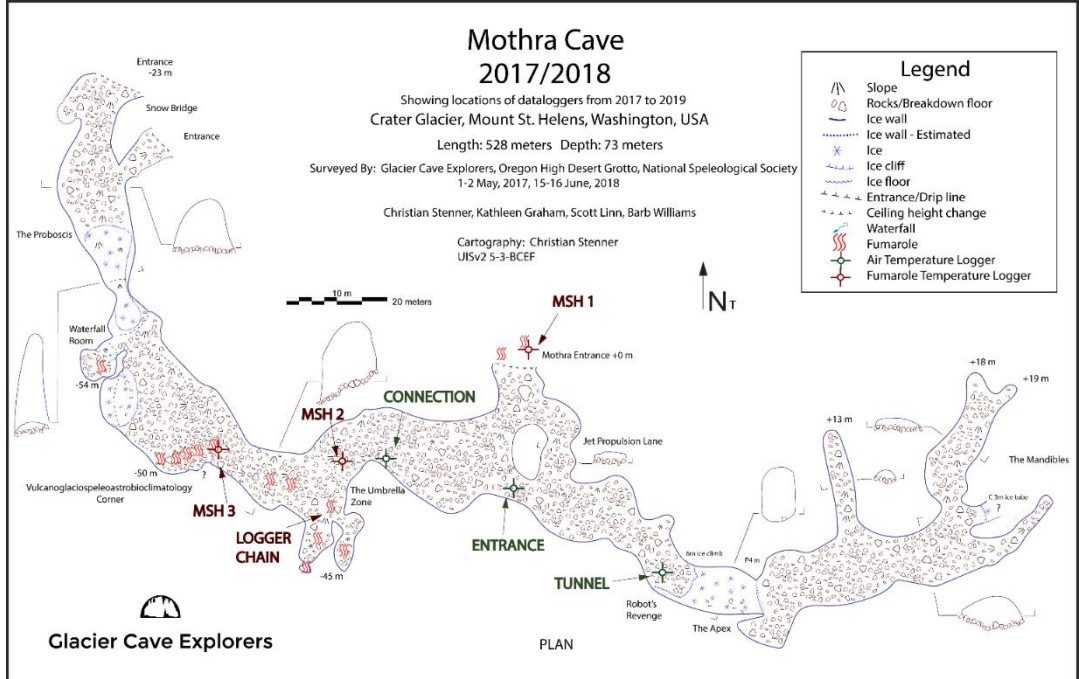

**Figure 4:** Map of Mothra Cave with sensor locations (fumarole and air temperature loggers) from 2017 to 2019. The cave morphology
reveals the situation between 2017 and 2018 (the first survey in 2017 was not complete and had to be finished in 2018).





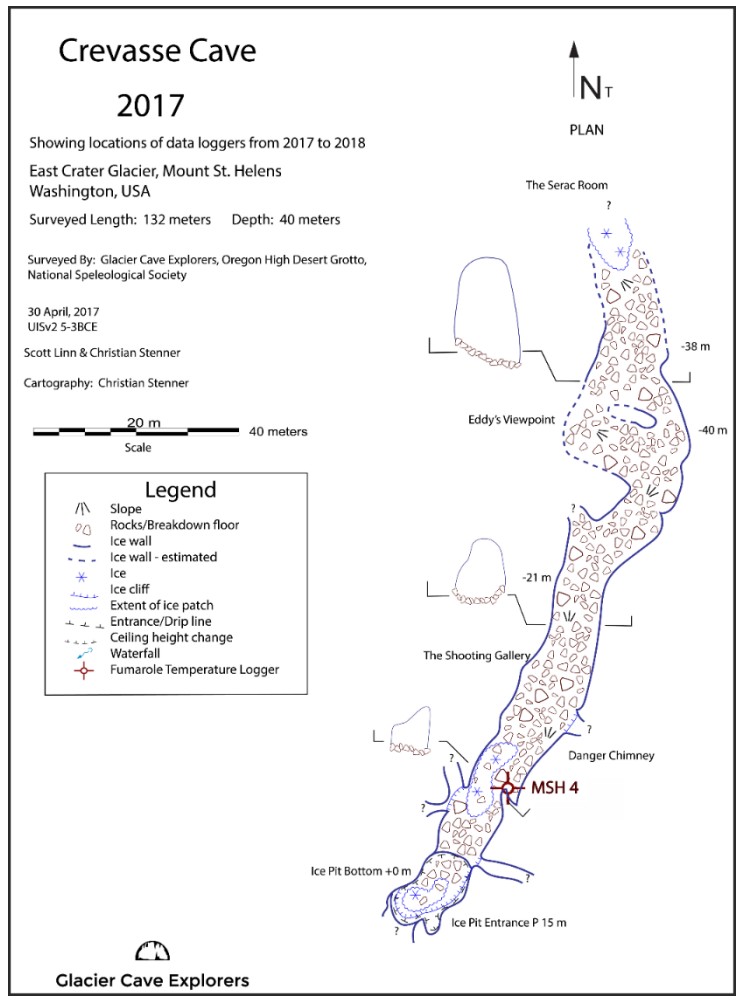

**Figure 5:** Map of Crevasse Cave with sensor location (fumarole temperature logger) from 2017 to 2018. The cave morphology reveals the situation in 2017.

The survey methods evolved during the project. Data in 2014 generated with the DistoX were manually recorded, with passage cross sections hand drawn at key stations. Passage dimensions were recorded in four directions at each permanent station. Survey data from 2017 onward were generated using the splay method with DistoX2 which includes more passage dimension measurements. The DistoX2 communicated with a Dell Axim X51 PDA and PocketTopo cave survey software or Samsung Galaxy Note 4 or similar Android OS devices and TopoDroid software. Challenges were given by interference of the DistoX

laser from water spray, fog, mist, or other obstructions, occasionally arising in the caves. In periods of limited visibility, splay measurements and detail were not possible. In those cases, basic passage measurements were estimated in four directions from the fixed stations. Other challenges included glacier movement and falling rocks which made it impossible to relocate some marked stations from previous surveys.



### 3.1.2 Data processing and cartography

Post processing of survey data was conducted in COMPASS cave survey project management software. Data was corrected for annual magnetic declination variance. Georeferenced stations input within COMPASS provided references for cartography over multiple survey years. The software generates statistics including length, depth, and volume. Due to magnetic interference from volcanic rock it was expected that some survey measurements generated by DistoX2 could be erroneous. These shots were mitigated by relying on loop closure capability of the software and GPS referenced stations where possible. COMPASS

makes this correction via the least square method (Schmidt and Schelleng, 1970). Finally, .kml files for ArcGis were generated, 3D visualizations in CaveXO software were produced, and corrected line plots and sketch data from software or manual recordings were used in Adobe Illustrator to complete final cartographic plan diagrams of each cave. With the use of historical imagery of the crater to identify the previous dimension of the crater and the location of the rock-ice-interface, it was also possible to calculate growth rates and the evolution of certain caves over the last years.

**3.2 Set up of climatologic measurement devices and data processing**

Two caves were subjected to climatologic studies, most inside Mothra Cave, and to a minor extent in Crevasse Cave. Between 2017 and 2019 GeoPrecision wireless data sensors (M-Log5W-CABLE) were placed in the caves to measure air and fumarole temperatures, adjusted to measurement intervals of five minutes. Locations were chosen to cover a large area inside each cave system, or to investigate interesting areas from a climatologic point of view. The installation of sensors and upload of data

cover the period from May to June as this is the safest time for fieldwork. In an analogous manner, a GeoPrecision sensor chain (digital thermistor string) of 20 to 30 sensors which are attached to the chain at intervals of one meter was installed to generate air and fumarole temperature profiles. In contrast to single data loggers, the chain was left in the caves during each expedition only, comprising three to five days. The chain was placed in an area with high fumarolic and individual loggers were positioned at fumaroles where possible. To detect hotspots inside the caves or outside on snow-free areas of the lava

dome a thermal camera (InfraTec VarioCAM HiRes) view was usually observed before placing any sensors. The direction and velocity of air flow was visualized by using orange smoke torches. Each implementation was documented by photography.

### 4 Results

### 4.1 Cave survey and general cave morphology

We surveyed 10 caves with a combined length of 2,335.7 m in detail, the three most significant ones reaching more than 400 m

each. Surveyed lengths ranged from 33.5 to 539.3 m. Depths varied between 7.7 and 65.2 m (Table 1). Glaciovolcanic caves on Mount St. Helens share many characteristics. The majority have formed in proximity to the 2004-2008 lava dome with passages trending parallel to the dome periphery (Fig. 2). As they have mostly developed near the lateral contact between ice and rock (Fig. 6), they can be characterized as marginal caves. Entrances usually exist along the interface of ice and the dome and passages descend along its slope at angles of +/- 30-40 degrees although vertical entrances are also observed. Most caves





exhibit two or more entrances, exceptions are two of the smaller caves, The Cloaca and Gabara Cave. Entrance elevations
range from 2,098 (The Igloo) to 2,256 m a.s.l. (Mothra Cave). Main passages tend to be horizontal and circumferentially
parallel to the ice-rock interface, large enough to be traversable by humans, with vertical sides and convex ceilings. All caves
exhibited large scalloping in the ice walls and ceilings. Some featured scalloped ice floors mirroring those features. Occasional
cryospeleothems were observed, including ice stalactites and stalagmites, and were associated with infiltrations of water.
However, the drainage of superficial water streams was not observed in the caves. Ambient air $CO_2$ measured $< 0.3$ %. Rocks
embedded in the glacier reveal themselves in the cave ceilings and walls through ablation. This creates a hazard present in
many of the caves. Various cave walls also contain diverse tephra layers. Cave characteristics are illustrated in figures 7a-f.

**Table 1**: Summary of caves and cave statistics showing the results from surveys done between 2014 and 2019. Last year of each cave survey
is indicated in figure 2. Cave statistics were generated with COMPASS.

| Cave | Included Length (m) | Horizontal Length (m) | Cave Depth (m) | Cave Volume (m³) |
|------|---------------------|------------------------|----------------|-------------------|
| **Crevasse Cave** | 275.7 | 261.8 | 56.4 | 27 307.4 |
| **Gabara** | 62.4 | 55.9 | 18.5 | 727.5 |
| **Ghidora** | 433.9 | 394.0 | 29.3 | 21 025.0 |
| **Godzilla Hole** | 176.4 | 163.3 | 41.3 | 10 662.2 |
| **Hedorah** | 98.6 | 87.4 | 13.4 | 6 542.8 |
| **Minilla** | 54.0 | 50.3 | 8.4 | 614.2 |
| **Mothra** | 539.3 | 502.8 | 60.7 | 40 903.5 |
| **Rodan** | 470.6 | 446.7 | 65.2 | 33 961.2 |
| **The Cloaca** | 33.5 | 25.2 | 10.3 | 285.9 |
| **The Igloo** | 191.3 | 187.2 | 7.7 | 4 323.4 |
| **Total** | **2 335.7** | **2 174.6** | - | **146 353.1** |

**Included Length**: This is the included slope length of all the surveys processed. Slope length is the sum of all the tape lengths in the cave.
It is the distance that you move through the cave, both horizontally and vertically.

**Horizontal Length**: This is the included horizontal length of all the surveys processed. Horizontal length is the length of the cave when it
is flattened into a horizontal plane. Horizontal length includes no vertical component.

**Cave Depth**: This is the absolute vertical distance between the highest and lowest points in the survey. It includes no horizontal movement.

**Cave Volume**: This statistic gives the volume of the cave surveys processed. It is based on the passage Left, Right, Up and Down
dimensions. Surveys that are missing LRUD's for part or all of the data will give inaccurate volume calculations.



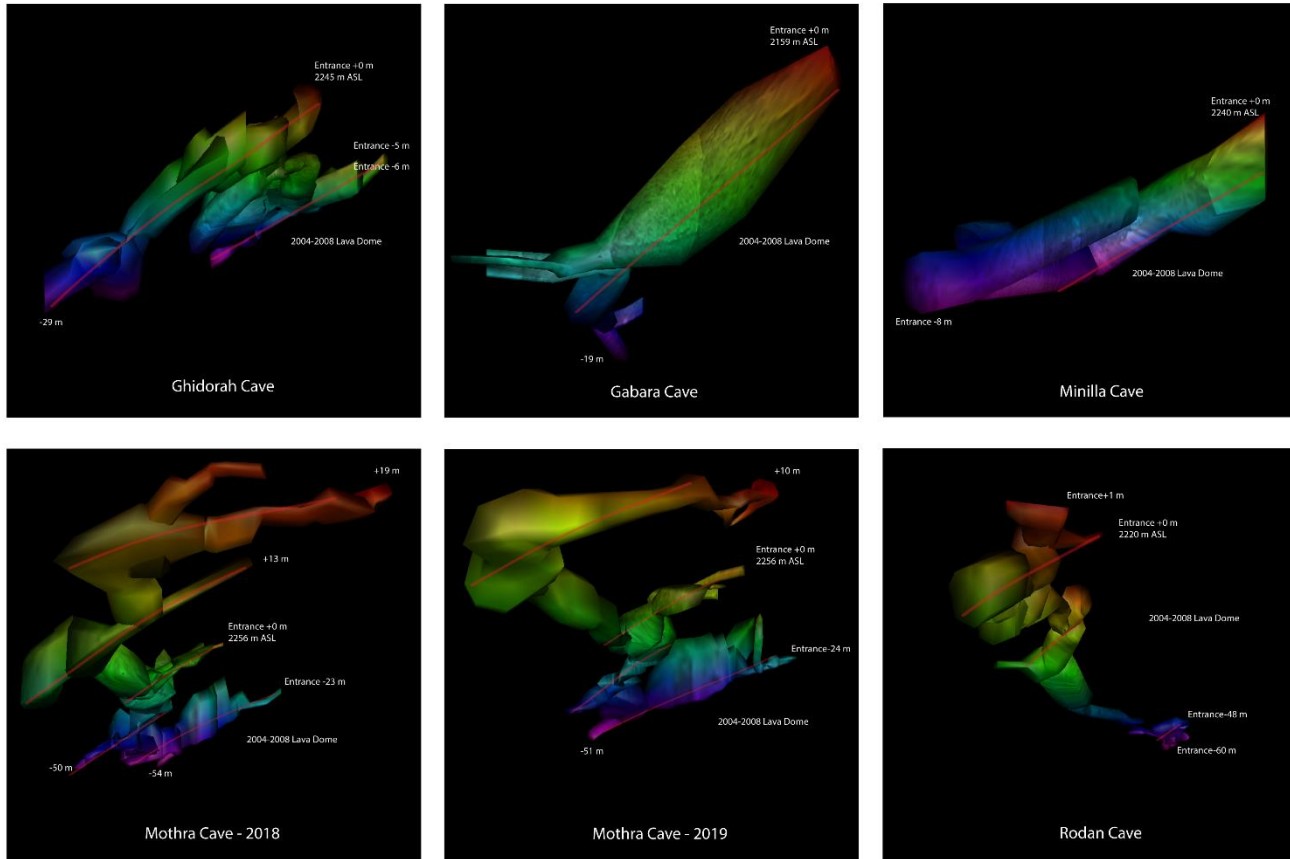

**Figure 6:** Profile views of selected Mount St. Helens caves. Profiles are rotated so that entrances along the 2004-2008 lava dome are to the right of the images. These caves show the prevailing complex morphology but also a pattern of higher elevation entrances along the rock-ice interface against the dome. Color range indicates relative passage depth (red: highest elevation, purple: lowest elevation).





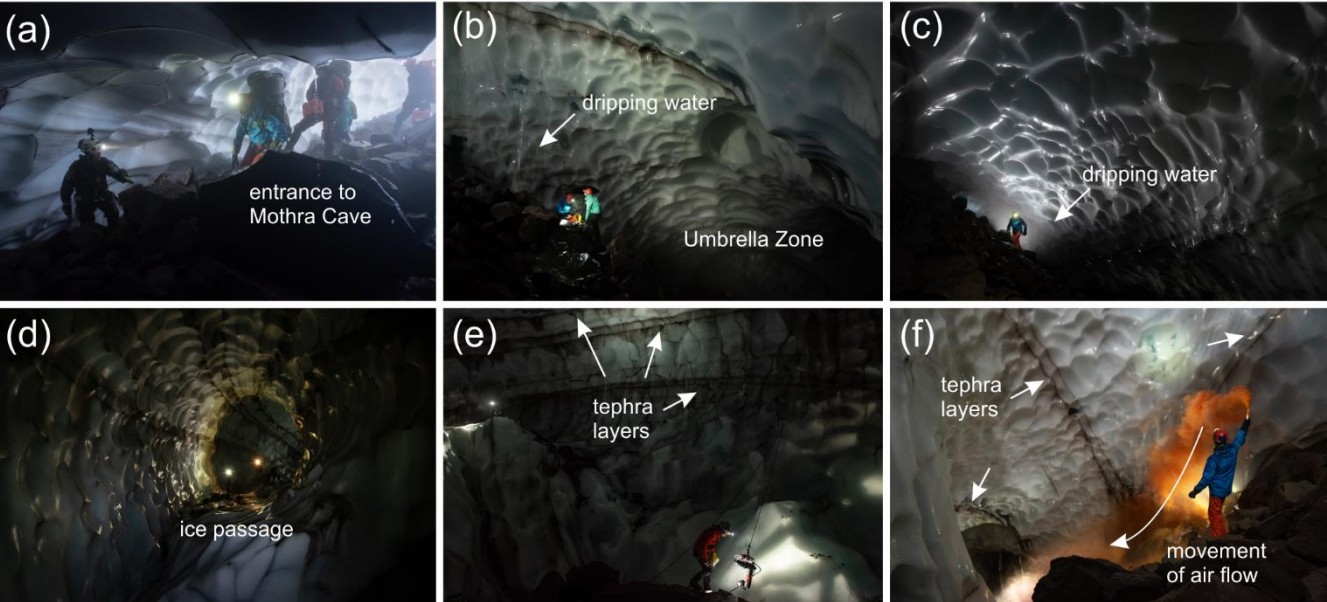

**Figure 7:** View inside Mothra Cave. **(a)** Entrance to Mothra Cave, large enough to be traversable by humans. **(b)** The Umbrella Zone indicating one of the wettest parts of the cave. Water is dripping from the ceiling. **(c)** The Umbrella Zone seen from another perspective. **(d)** Elliptical ice passage revealing the dimensions of the cave. **(e)** Numerous tephra layers. **(f)** Tephra layers and movement of air flow which is made visible with a customary orange smoke torch. All images: Guth, E.

### 4.1.1 Mothra Cave

Mothra Cave was the longest of all the Mount St. Helens caves after initial surveys were completed from 2017 to 2018. This cave has two entrances which exist at the rock-ice interface on the south slope of the 2004-2008 lava dome. Morphology characteristics include large passages, steeply sloping floors of debris and two large rooms, 20 x 20 x 13 m and one of 14 x 21 x 16 m. A rift-like passage 7 m wide separates them with a maximum chamber height of 28 m. An exceptional feature is an elliptical ice passage 6 m above the debris floor which extends for nearly 15 m (Fig. 7d). Dendritic passages trend upward beyond this feature as the cave passages approach the apex of Crater Glacier. The Waterfall Room and The Umbrella Zone (Fig. 7b) feature small waterfall infiltrations from the ceilings. From resurveys of Mothra Cave, the latest in 2019, morphological changes were apparent. The Waterfall Room no longer existed along with changes to the south wall and the dendritic passages on the east branch. The master passage predominantly remained stable with changes in passage height being most apparent, and a lateral shift of nearly 10 m to The Umbrella Zone. Those changes are documented in two maps, one illustrating the survey in 2017/2018 (Fig. 4) and the other in 2019 (A1). Areas with prominent fumarolic activity have been observed in the cave over multiple survey years and were in the two largest rooms. Although single fumarole locations have changed or disappeared from one year to the other, the concentration of increased activity in certain areas was stationary. Based on historical imagery of the crater, we calculated a growth of more than 240 % in length and 530 % in volume for the



cave from 2012 to 2018 (Fig. 8). Arial photos from June and July 2020 indicate further expansion via a new opening located on top of the tall chamber on the west side of the cave.

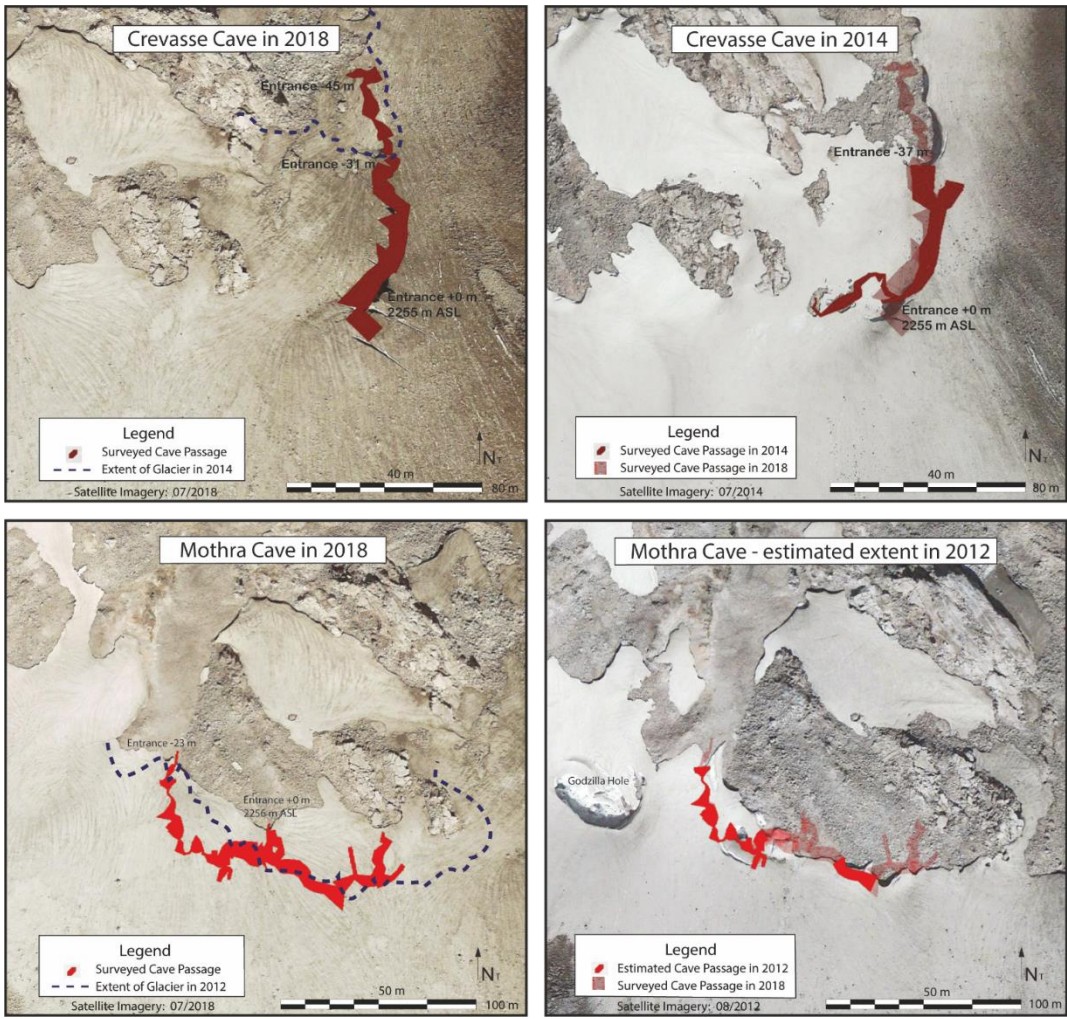

**Figure 8:** Location and extension of Crevasse Cave in 2018 compared to cave passages surveyed in 2014 (top). Comparison of Mothra Cave as of 2018 and projection of the cave passages onto 2012 imagery indicating the movement of the rock-ice interface to higher elevations along the 2004-2008 lava dome and the increase in cave passages that would have formed since 2012 (bottom). Both, Crevasse Cave and Mothra Cave, demonstrate the rapid rate of cave genesis. The approximate extension of Crater Glacier in 2014 for Crevasse Cave and 2012 for Mothra Cave is illustrated onto 2018 imagery. Satellite imagery originate from Google Earth. © Google Earth.


**4.1.2 Crevasse Cave**

Crevasse Cave was discovered in 2014. The cave was entered through a 5 x 7 m opening to a nearly vertical shaft approximately 20 m deep which led to the cave floor. The shaft intersected a master passage with a breakdown floor trending southwest-





northeast. The southwest passage was explored to a dendritic series of chambers with three skylights. The northeast branch
was explored to a horizontally oriented entrance at the rock-ice interface (B1). Resurveys of Crevasse Cave in 2017 (Fig 5),
2018 (B2) and a brief exploration in 2019 revealed distinct changes within the cave. In 2018 a new skylight entrance had
formed located alongside the lava dome north of the shaft entrance at a lower elevation. This connected to the master passage
which extended further north to another new entrance along the east side of the 2004-2008 lava dome. This, in turn, connected
to the known master passage surveyed in 2014. Within the master passage a new sloping ice floor formed and the passages,
similar in size, had shifted west. Fumarolic activity in Crevasse Cave was not surveyed in detail during the first explorations
in 2014 and 2017. The survey in 2018 included the first detailed mapping of fumarole locations. Fumaroles were present in
the center and new northern part of the cave. Extrapolated from survey data, the northward-trending main passage revealed a
growth of more than 100 % in length and roughly 250 % in volume between 2014 and 2018 (Fig. 8).

### 4.1.3 Features of remaining caves

Centrally located in between the crater rim and the lava dome, the Godzilla Hole was the first cave investigated in 2014. This
cave has not been entered since initial exploration in 2014 since the entrance pit, a 10 x 20 m opening, was no longer present.
From 2017 to 2020 a closed depression remains in the glacier at its former location. The Igloo is the northernmost cave in the
crater and was discovered in 2014. It is contained completely in firn ice 8 m in depth. Its dynamic passages reform every
season but a distinctive persistent central hemispherical chamber centered around a fumarole vent gives the cave its name. The
remaining caves on Mount St. Helens were discovered in 2018 or 2019. Smaller caves formed in firn ice include Minilla, The
Cloaca, and Gabara. All of them are marginal caves sharing similar characteristics. Rodan Cave is the second longest cave in
the crater and is the deepest. It consists of a large master passage wrapping circumferentially around the southwest side of the
2004-2008 lava dome. The master passage is intersected by large crevasses and some hemispherical rooms. In the western
portion of Rodan Cave this laterally oriented master passage connects a section with upward trending morphology (Fig. 9).
Hedorah Cave, located east of Rodan Cave, exists near a group of fumaroles. It has five unique entrances spaced closely
together and oriented towards the fumarole field. Ghidorah Cave represents the last distinctive cave of the crater and most
closely resembles a dendritic passage network which is common in karstic caves. Its three north facing entrances are at the
rock-ice interface on the south slope of the 2004-2008 lava dome.



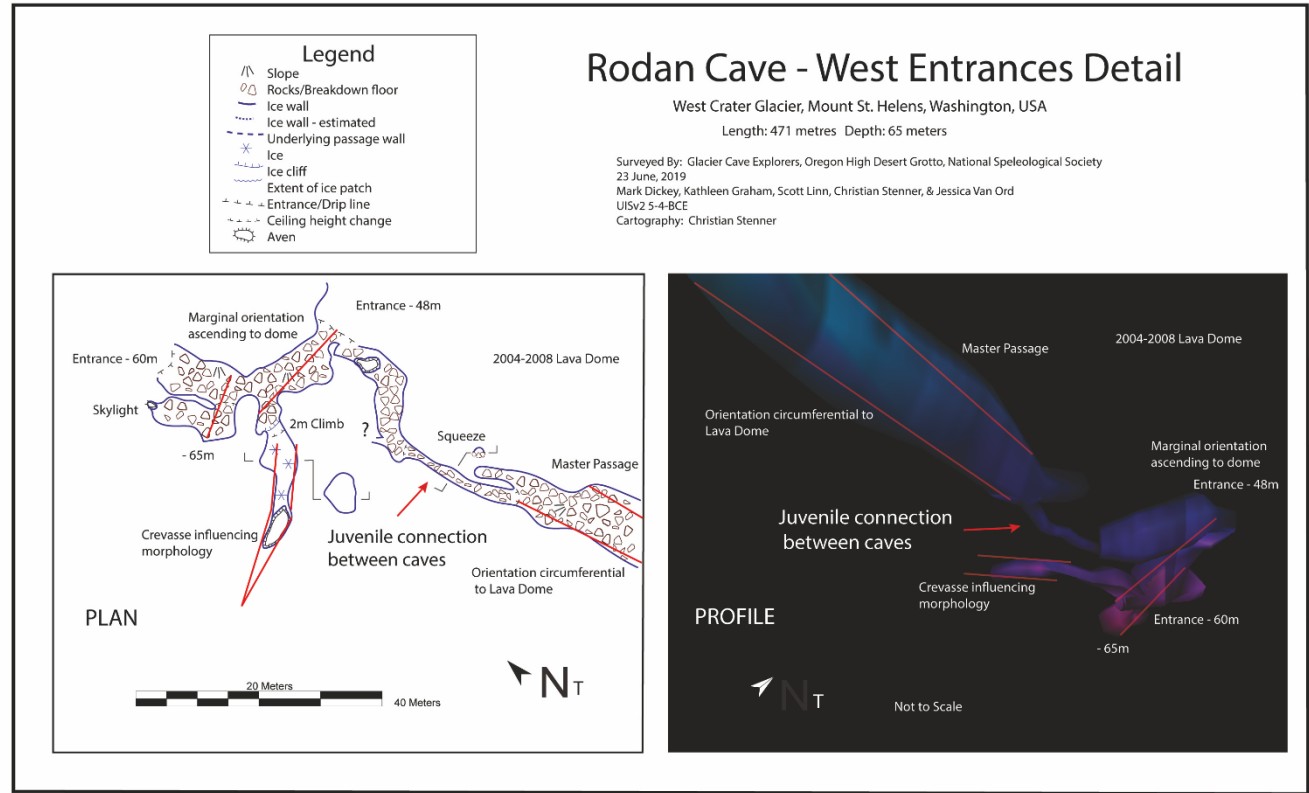

**Figure 9:** Map of Rodan Cave (west entrances) and profile view. A laterally orientated master passage connects a section with upward trending morphology. Rodan Cave is supposed to be a juvenile cave system where the formation of connecting cave passages has started recently. Continuing heat output will probably lead to further expanding.

## 4.2 Air temperature measurements

Air temperatures in the entrance area (-16 m) range from 8.6°C recorded in May 2017 to -7.2°C in November 2017. The temperature curve reveals a clear seasonality with distinct minimum temperatures from October to November 2017 and November to December 2018. Temperatures above 0°C prevail during the majority of the year. Sub-zero temperatures emerge at the earliest in the middle of September and continue throughout November or December. Exceptions of sub-zero temperatures occur in other months but usually do not fall below -1°C. Maximum temperatures recur annually with the end of winter throughout springtime from February to May (Fig. 10a). The connection passage (-24 m) reveals a maximum temperature in May 2017 of 10.6°C. A minimum of -5.5°C was recorded in December 2018. Like the entrance area, the connection passage exhibits seasonal patterns with temperatures prevailing above 0°C for most of the year, excluding the months from September to December. Contrasting the connection passage, a distinct minimum could not be observed in winter 2017 (Fig. 10b). The tunnel area (-11 m) represents the highest of the three locations. Like the entrance area and the connection passage, the temperature curve is subjected to periodicity. Maximum temperatures of 4.4°C were recorded in May 2017,



minimum temperatures of -3.4°C in November 2017. From December 2018 until the end of recording the curve predominantly remains around 0°C with marginal divergence in positive and negative directions not exceeding a span of 1°C (Fig. 10c). We assume that the data logger was frozen in ice and reveals stages of thawing and refreezing cycles. Amplitude scaling and offset

260    translation indicate that temperature profiles at all three locations follow the same pattern during the year (Fig. 10d), and also mirror diurnal trends with temperatures increasing at noon and afternoon and decreasing in the evening (Fig. 11). Over the course of one day temperatures in the entrance area reveal most distinct amplitudes (3.6°C), followed by the connection passage (2.1°C) and the tunnel area (0.9°C).

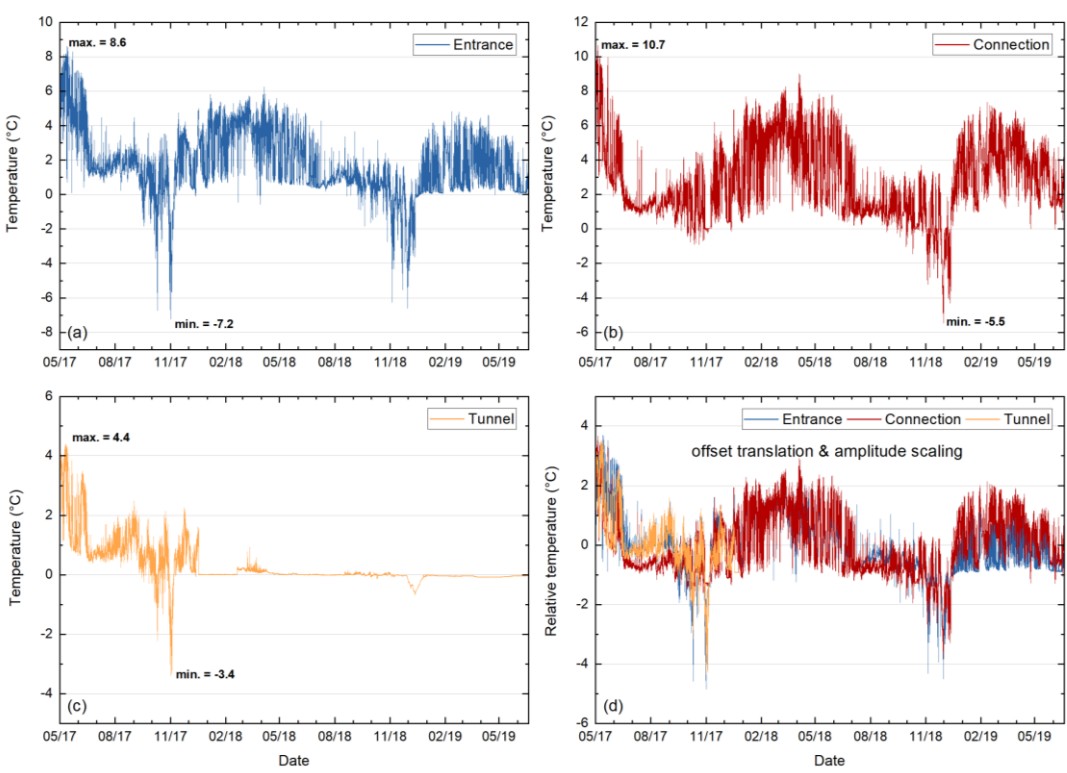

265

**Figure 10:** Air temperature profiles from three different locations inside Mothra Cave. **(a)** Entrance. **(b)** Connection. **(c)** Tunnel. **(d)** Entrance, Connection, and Tunnel after offset translation and amplitude scaling. Air temperature after offset translation and amplitude scaling is indicated as the relative temperature.

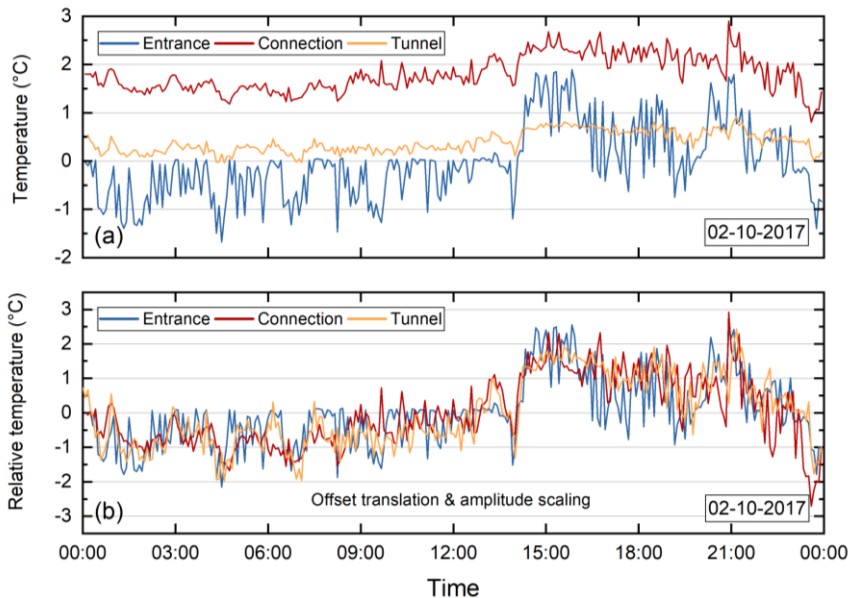

**Figure 11:** Air temperature profiles from three different locations inside Mothra Cave in the course of one day (2 October 2017). **(a)** Air temperature profiles from the entrance area, the connection passage, and the tunnel area. **(b)** Air temperature profiles after offset translation and amplitude scaling. Air temperature after offset translation and amplitude scaling is indicated as the relative temperature.

A correlation between monthly mean temperatures inside Mothra Cave and the average snow depth per month at June Lake exists. The correlation coefficient after Pearson was calculated as 0.77 for the entrance area and 0.91 for the connection passage. Coefficients after Spearman analogue range between 0.73 and 0.87 (Fig. 12). Snow depths represent a periodicity with snow accumulation between November and June and maxima in March and April, conform with maxima of temperatures. One exception is represented by spring 2019 where temperature maxima arise the month before and/or after the snow maximum. Equally to the entrance area and the connection passage, the tunnel area mirrors a correlation during the spring and summer season in 2017 but data were not considered for further calculation due to interference of the statistical series.

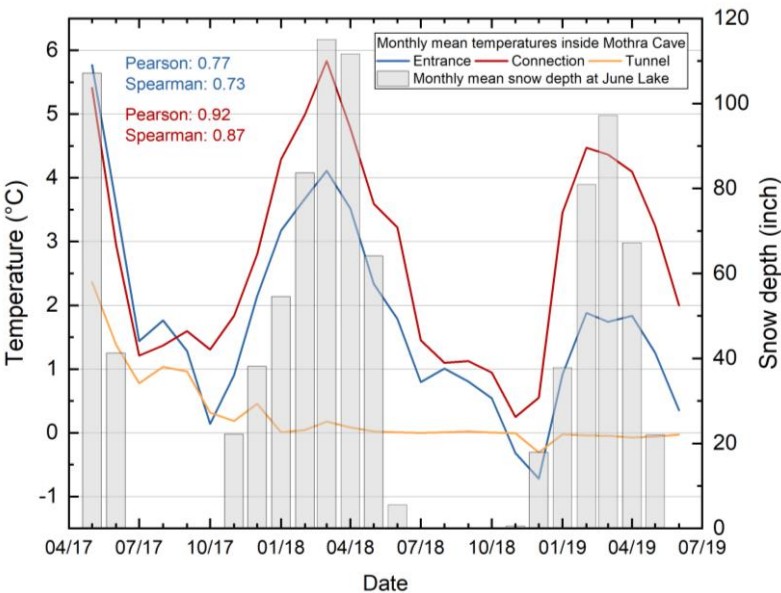

**Figure 12:** Monthly mean temperatures versus monthly mean snow depth at June Lake, Skamania County, south side of Mount St. Helens at an elevation of nearly 1,050 m. Correlations after Pearson and Spearman illustrate the relation between monthly mean air temperatures in the entrance area and the connection passage versus monthly mean snow depth at June Lake. A correlation for the tunnel area was not calculated since the data logger probably got frozen into glacial ice after the first year. Snow depth data was provided by the NRCS National Water and Climate Center website, online accessible (https://wcc.sc.egov.usda.gov/nwcc/site?sitenum=553&state=wa).

## 4.3 Observation of ventilation effects and air flow movement

We were able to notice ventilation effects and air flow movement. Strongest ventilation occurred near entrances, minor effects were experienced at higher elevations inside the caves and areas further away from the entrances. The release of orange smoke inside Mothra Cave (Fig. 7f) and Crevasse Cave in June 2018 similarly visualized the movement of air flow. A downward directed movement, following the cave morphology and the steep slopes, was observed inside Mothra Cave. Moreover, the air was flowing deeper into the cave system directing to the western part of the cave. The smoke was set free in visual range of the main entrance (0 m) in between the data loggers of the entrance area and the connection passage. In Crevasse Cave the smoke equally was set free in one of the entrance areas. Similar to the situation inside Mothra Cave the smoke moved into the cave. At both cave systems a subsequent release of smoke through other entrances or openings was observed. Orange smoke at the surface was visible a few minutes after its release inside the caves.

## 4.4 Fumarole temperature measurements

Fumarole temperatures on Mount St. Helens show a great range of fluctuation with maximum temperatures of nearly 60°C and minimum temperatures sub-zero (Fig. 13a). MSH 1 is the fumarole with the strongest fluctuation from -10.0 to 60.1°C. This fumarole shows a general temperature decrease over time and a clear seasonality. MSH 3 reveals the least extensive fluctuations and temperatures range from 1.4 to 46.6°C. MSH 2 and MSH 4 range from -0.1 to 57.4 and from 0.9 to 57.1°C.



Time series for MSH 2, MSH 3, and MSH 4 are shorter than for MSH 1 but data do not nearly indicate a seasonality as
305   distinctive as observed for MSH 1. Another time series of fumarole temperatures exists from Mount Rainier. Temperatures
here are less variable and do not show such a strong range of temperatures with some outliers being exceptions (Fig. 13b).
Temperatures are constantly higher and usually do not fall below 40°C. A minimum temperature appears in August 2015 with
20.8°C and include one of the outliers being observed. Maximum temperatures with 60.4°C are similar to those measured on
Mount St. Helens. On Mount Rainier temperatures rather reveal changes from one year to another than indicating seasonal
310   variability within one single year. Similar trends could at least be observed at MTR 1 and MTR 2 whereas some correlations
of outliers also apply to MTR 3. To some extent, this fumarole reveals a reverse temperature profile in contrast to MTR 1 and
MTR 2. Temperatures increase at MTR 3 when decreasing at MTR 1 and MTR 2 and vice versa. In a confined space fumarole
(and air) temperatures range from 0.7 to 45.7°C and reveal a great variability according to the degree of dispersion. Although
individual loggers depict similar patterns (e.g. logger 3 and 4; 23 to 26), other loggers positioned side by side indicate
315   completely different temperature distributions (e.g. logger 4 and 5; 20 and 21). The boxplots demonstrate disparity between
individual loggers (Fig. 14). Individual fumaroles on top of the 2004-2008 lava dome revealed temperatures around 90°C and
were detected with the thermal camera.

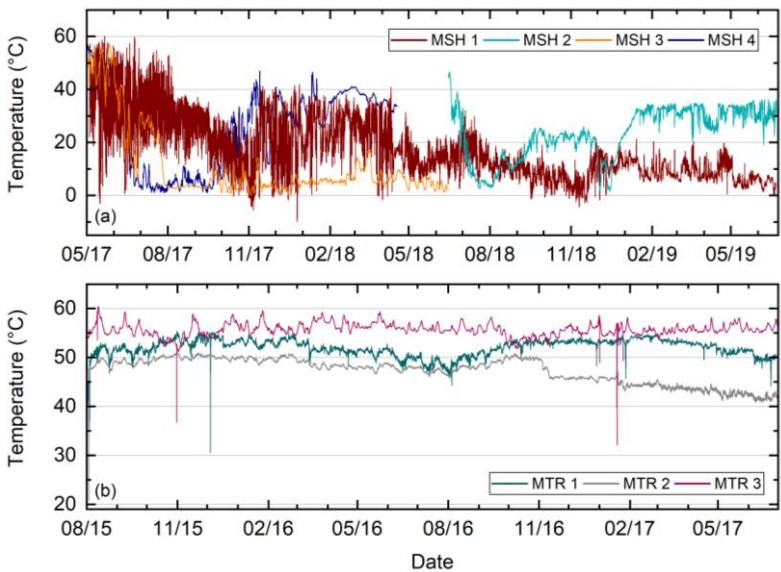

320   **Figure 13:** Fumarole temperatures from Mount St. Helens and Mount Rainier. **(a)** Fumarole temperatures from three different locations
inside Mothra Cave and one location inside Crevasse Cave, Mount St. Helens, between May 2017 and June 2019 (Figs. 4 and 5). **(b)**
Fumarole temperatures from three different locations inside the East Crater Fumarole Cave, Mount Rainier, between August 2015 and July
2017.

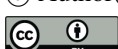



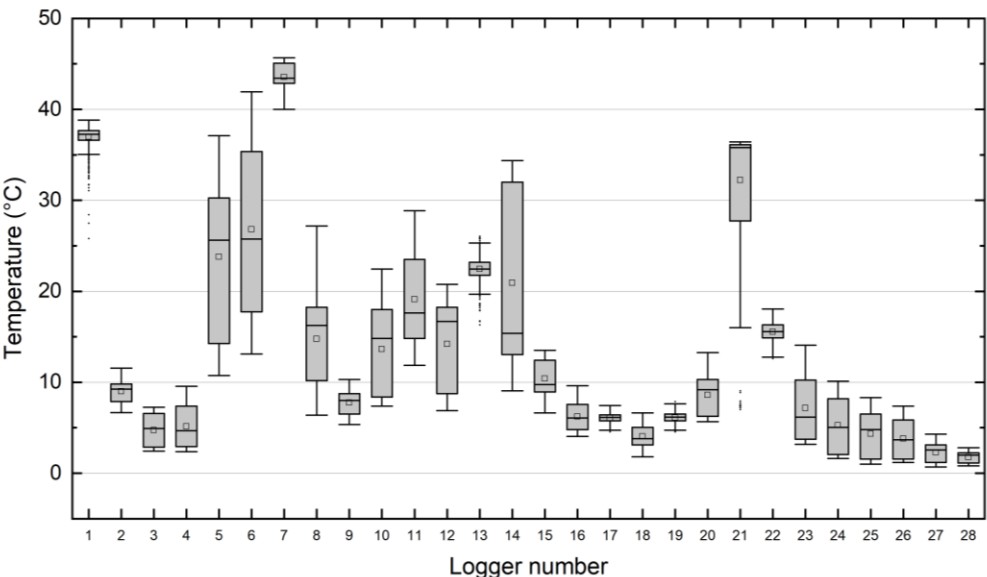

**Figure 14**: Boxplot diagram representing data from a logger chain with 28 sensors which was placed inside Mothra Cave (Fig. 4). Data represent measurements from 15-18 June 2018.

## 5 Discussion

### 5.1 Fumarolic activity and cave hydrology

Fumaroles are concentrated around the 2004-2008 lava dome which consists of young lava and is still hot at depth and the main driving force of fumaroles. The location of cave systems correlates with fumaroles, which particularly embed the area south of the dome and some parts of the eastern and western arm of the glacier. Other fumarolic activity has been observed near the 1980-1986 lava dome correlating with the location of The Igloo. Apart from the area of glaciovolcanic caves, fumaroles also appear on top of the recent dome and near cave entrances. Although cave systems appear all around the dome, closer investigations show that there are locations in most of the caves where fumaroles are concentrated. Hemispherical rooms, increased ablation, and the absence of ice on the floor are typical indicators. The distribution of fumaroles around the dome indicate that volcanic heat of the young dome is responsible for cave systems to evolve. The formation at individual concentrated spots and varying temperatures in turn is controlled by cracks and fissures. Thus, the origin of each cave is closely related to the abundance of fumaroles as they determine the onset of formation and control further genesis.

Fumarole temperatures do not exceed 60.1°C. This is enough heat for cave systems to form. A wide range of temperatures is revealed by single fumaroles not only annually but also in the timeframe of a few days. They primarily fluctuate slightly above-zero up to 40°C. Higher temperatures are locally and temporarily limited. Some fumaroles even revealed temperatures sub-zero before continuing their heat output once more. We assume that ablation of cave ceilings and walls as well as rainfall are



the most determining forces. Equally important is the nature of the crater floor. Ablation is constantly given through the output of heat and therefore self-energizing where fumaroles are concentrated in a cave. Precipitation at Mount St. Helens appears both as snow and rainfall due to its comparatively low elevation. The crater floor usually contains large boulders mixed with some smaller rocks. The appearance of chemically or physically weathered material is limited since most of the cave systems evolved on recently erupted material according to the 2004-2008 dome growth. This results in a permeable underground.

Furthermore, the crater floor is characterized by steep slopes. We were able to enter into the Crater Glacier to depths of more than 65 m and did not detect streams or pooling in any of the caves. We conclude that the continuous presence of rainfall and glacial ablation in combination with the features of the crater floor essentially affect fumarole temperatures and cause this strong fluctuation. Cracks and fissures are constantly influenced by infiltrating water. We observed different situations in other glaciovolcanic cave systems. This once more indicates the significance of the hydrologic situation.


Mount Hood's cave systems feature several streams and cascades. The caves contain fog and reveal a strong humidity. Fumaroles as observed on Mount St. Helens usually do not exist and rather arise as hot springs (Pflitsch et al., 2017). Intensified superficial water runoff has been observed. Altered volcanic material and loamy gravel exist inside the caves. Typical situations experienced on Mount Hood are illustrated in figures 15a-f. The crater floor on the summit of Mount Rainier is characterized

by various clays which result from strong hydrothermal alteration (Zimbelmann, 1996). The morphology is featured by natural depressions and is mainly controlled by the location inside the craters and the abundance of glacial ice plugs. Unique subglacial lakes are able to form (Figs. 16a-f). Since the summit craters are located at high elevations, snow precipitation dominates. These contrasts to Mount St. Helens are revealed by fumarole temperatures. Although we equally observed variations over time, this apparently occurs at a much smaller scale. Fumaroles also remain at constantly high temperatures and fluctuate

roughly between 40°C and 60°C. Some exceptions with abrupt temperature decreases may be related to sudden and strong rainfall events, increased ablation, or temporary changes and disturbances of the hydrothermal system. Fumaroles furthermore depict a kind of dependency and show similar trends over a period of two years although a distinct seasonality is missing (Florea et al., 2020, in review). Fumaroles on Mount St. Helens seem to follow independent pathways in so far, that they do not reveal concordant trends. Although a distinct seasonality cannot be confirmed for the fumaroles in general, at least MSH

1 in the entrance area mirrors seasonal variations. We therefore conclude that fumaroles are affected by seasonality but this applies more to the areas aside from the caves. Explicit reactions of fumaroles inside seem to be absent or hardly noticeable.





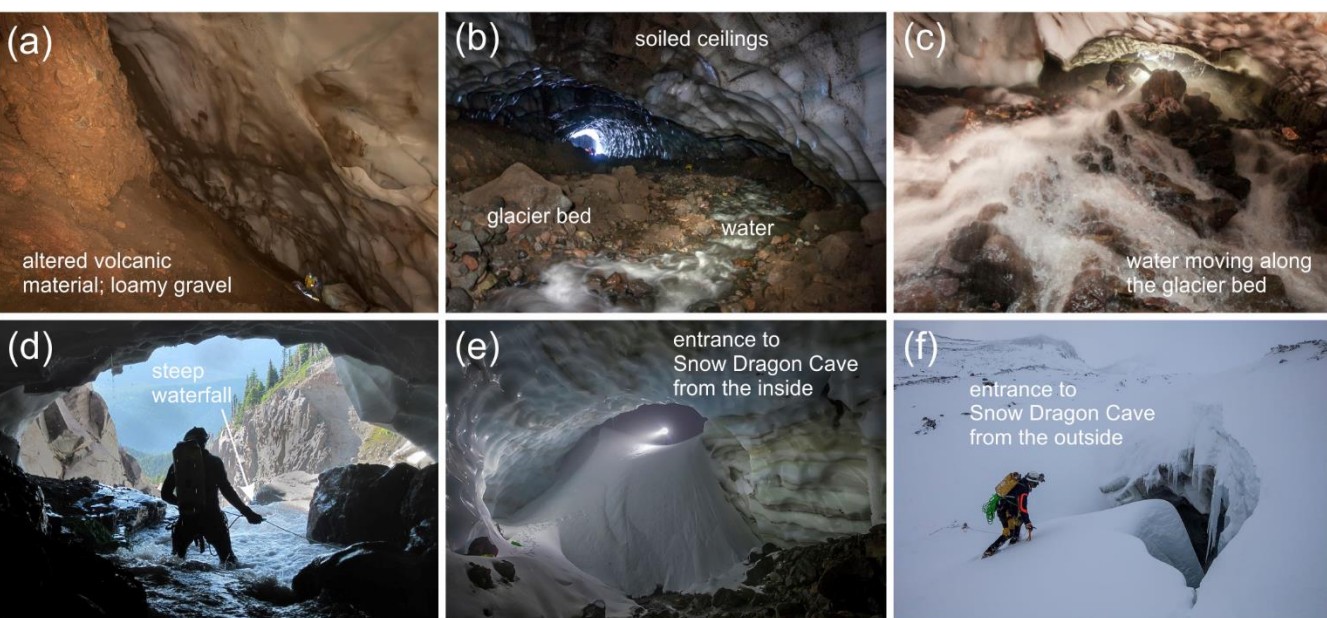

**Figure 15:** Sandy Glacier Cave System, Mount Hood. **(a)** A large room connected to the upper end of Pure Imagination Cave showing altered volcanic material, June 2016. **(b)** Upper section of Pure Imagination Cave, July 2018. The glacier bed is characterized by superficial water. The ceilings contain large amounts of volcanic material. **(c)** Sudden flood in Pure Imagination Cave, July 2017. **(d)** Snow Dragon Cave with a steep waterfall, August 2011. **(e)** Entrance to Snow Dragon Cave a few weeks before it sealed shut with snow, January 2012. **(f)** Entrance to Snow Dragon Cave from the outside, January 2012. All images: McGregor, B.







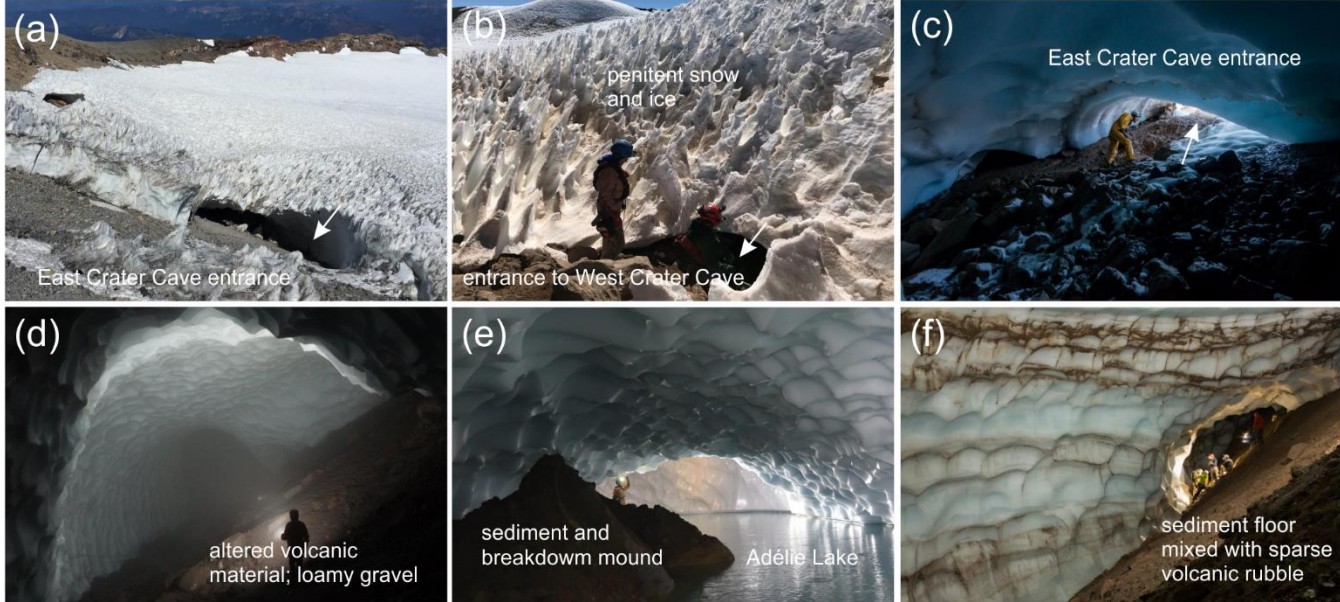

**Figure 16:** Mount Rainier, cave systems summit craters. **(a)** East Crater Cave entrances from near Columbia Crest, the summit of Mount
Rainier. View looking northeast. Image: Stenner, C. **(b)** Entrance to West Crater Cave. Penitentes visible on the glacier surface. View facing
east. Image: Stenner, C. **(c)** An East Crater Cave entrance from inside. Image: DeRuydts, F. **(d)** The Coliseum, East Crater Cave. An example
of the largest of cave passages in the East Crater. Sediment and volcanic breakdown floor and heavily scalloped ice walls. Image: Riggs, D.
**(e)** Adélie Lake in East Crater Cave. A sediment and breakdown mound to one side of the lake. Image: Wood, T. **(f)** The largest room in
West Crater Cave. Mostly sediment floor mixed with sparse volcanic rubble. $CO_2$ pools at the bottom of the chamber. Image: Riggs, D.

Walder et al., (2008) suggested that much of the rubble underneath Crater Glacier is likely to be ice free because of the

geothermal heat flow. Our observations turned out that this applies to most parts of the caves. An exception to this are

fumarole-free areas in the vicinity of entrances where occasional cold air can lead to ice sheets that are several centimeters

thick as well as icefall from ceiling and rime ice coatings. He also suggested that water that reaches the glacier bed probably

flows out of the crater through the rubble layer or downward into the volcano's groundwater system rather than moving along

the glacier bed. His suggestions additionally support our assumption that fumarolic activity is significantly controlled by cave

hydrology and the nature of the crater floor. Volcanic heat and fumaroles as the driving force initiate the onset of cave formation

before further processes start to arise.

**5.2 Ventilation effects and air temperature**

Strong ventilation effects inside Mothra Cave and Crevasse Cave have been observed and visualized with smoke. This situation

results from a complex cave morphology with several cave openings in different elevations causing strong chimney effects.

Continuous transitions from ascending warm air heated by fumaroles and gravitational cold air exist as long as cave entrances

are open and an exchange with external air is possible (Fig. 17). We assume this situation applies to most of the caves on

Mount St. Helens. Moreover, ventilation effects potentially control the amount of $CO_2$ being present in the caves. Although





the amount of gases being emitted from the magmatic system is most important, ventilation effects also control the subsequent distribution and accumulation within the caves. On Mount St. Helens we measured $CO_2 < 0.3$ %, probably indicating ventilation and the missing facility to accumulate. Dangerous amounts of $CO_2$ have been experienced in the west crater of Mount Rainier (Stenner et al., 2020, in preparation; Zimbelman et al., 2000). Compared to Mount St. Helens, ventilation effects here are likely to be less distinctive due to the different morphology, primarily minor elevation changes, and chimney effects

arising to a lesser extent. Moreover, the crater floor is less steep and impermeable, thus providing more possibilities for $CO_2$ to accumulate and to get trapped.

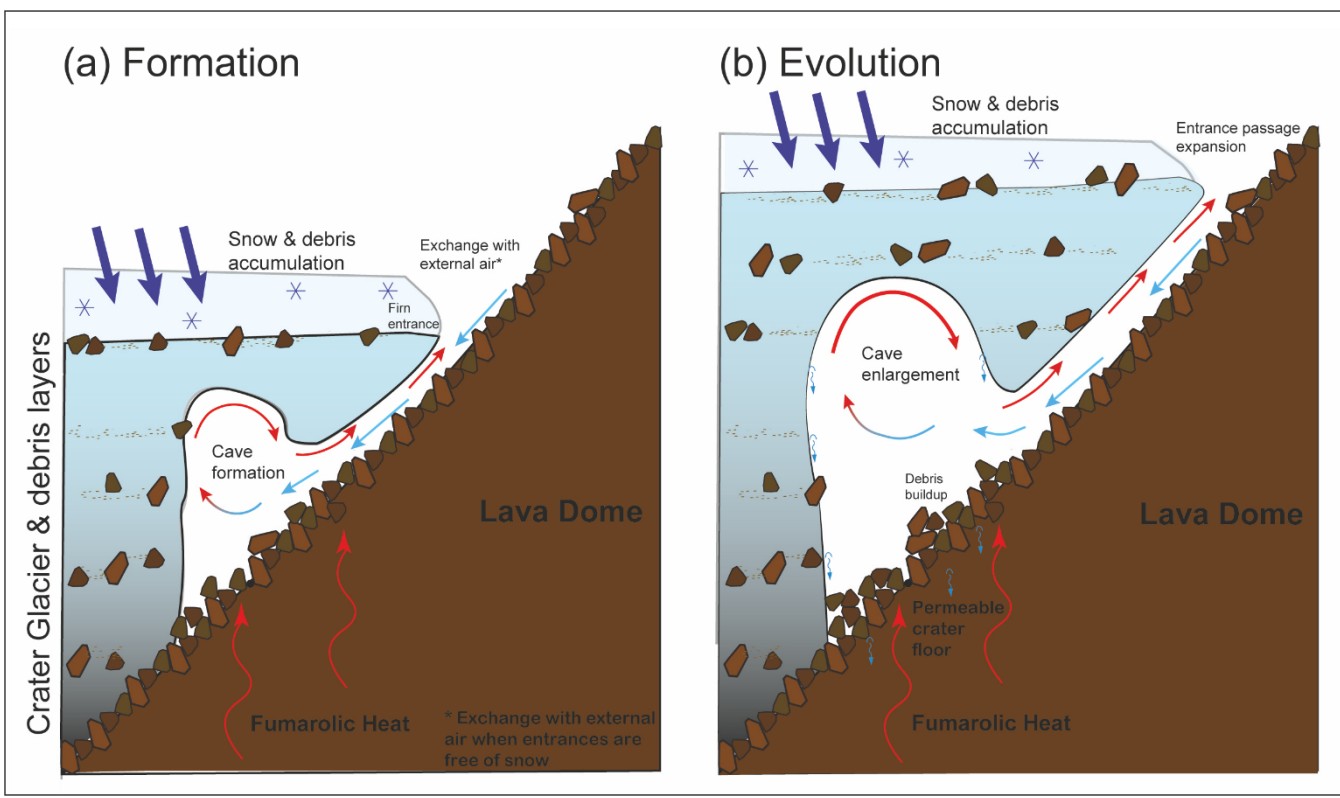

**Figure 17**: Future evolution of caves viewed in profile, using Mothra Cave as an example. Subglacial cavern enlargement continues laterally,
and vertically by way of passage enlargement and passage lengthening towards the lava dome with further glacial accumulation. **(a)** Onset of formation due to fumarolic heat. **(b)** Further evolution and cave enlargement. Exchange with external air arises as long as cave entrances are not sealed with snow. The high content of debris and the permeable crater floor are exceptional features of Crater Glacier.

Distinct relations between air temperatures and ventilation effects exists. Most important is the seasonality of air temperatures
on Mount St. Helens. Highest temperatures clearly correlate with major snowfall since entrances get sealed and warm air is trapped. This accelerates further melting and a self-energizing process arises. Based on this, warmest cave temperatures exist in winter and spring, coldest temperatures in summer and autumn. Similar situations are known from Mount Hood (Figs. 15 e-f) and the fumarole ice caves on Mount Rainier (Florea et al., 2020, in review; Pflitsch et al., 2017; Zimbelman et al., 2000).

In months without snow strong venting effects on Mount St. Helens influence the cave climate and reduce the concentration
of fumarolic heat. These effects were equally observed inside the former cave systems (Anderson and Vining, 1999). We
assume that the absence of ventilation effects during winter and spring and resulting melting processes from the inside
contribute much more to the transformation of the glacier than ablation processes at the surface in summer and autumn.

Temperature profiles at different locations inside Mothra Cave reveal a peculiarity and are subjected to the same seasonal
cycles though they are fluctuating within a different amplitude. As expected, the area nearest the entrance as well as the
connection passage are most influenced by seasonal changes. Those comprise relative elevations of -16 and -24 m where the
entrance represents a reference point with an elevation of 0 m. Another data logger placed inside the tunnel area, the highest
of all three sensor elevations at -11 m, recorded the smallest temperature amplitude. The morphology and different elevations
inside a cave system have distinct influence on the development of small-scale climatologic characteristics. During the year
but also during one single day temperatures in the connection passage are usually the warmest. This is the location closest to
concentrated fumarolic activity. Occasional warmer temperatures near the entrance may be related to minor and temporary
changes of fumarole degassing. The tunnel area also reacts to external factors, but to a much smaller extent. The absence of
concentrated heat in this part of the cave is visible through the appearance of an icefall and is moreover revealed by lower
monthly mean temperatures and the assumption that the data logger got frozen into glacial ice. Whereas the connection passage
has the highest temperatures for most of the year, the entrance area is much more sensitive to external changes as long there is
a direct exchange with the outside air and entrances are open. This means cave air temperatures develop in an analogue pattern
to outside air temperatures.

**5.3 Cave morphology and further evolution**

Ventilation effects, the distribution of air temperatures, the morphology of the crater floor, and volcanic heat interact in several
ways. This causes the formation of caves around the dome, internal morphology changes, and circumferential evolution like
the cave system on Mount Rainier. Fumaroles as the driving force primarily influence the evolution of caves and their
morphology. During our surveys distinct changes within cave systems were related to the appearance or disappearance of
fumaroles. Although individual fumaroles changed, we conclude that major fluid pathways must have remained. Minor
changes may be related to the growth or decay of cracks even though this could not be directly observed on the crater floor.
Distinct shifts related to changes of fumarolic heat have been observed for Crevasse Cave with passages now extending further
northwest (Fig. 8) and the disappearance of The Waterfall Room inside Mothra Cave (A1). Nevertheless, main cave passages
remained, indicating that major fluid pathways did not change. This can also be confirmed for The Igloo.

Observations and resurveys of the caves revealed significant dynamics and the trend to expand in the near future. Historical
imagery and the reconstruction of former cave dimensions (Mothra Cave) emphasize the immense growth during the last
decade. Arial photos from June and July 2020 moreover indicate further expansion. Crater glacier is growing and as the





accumulation of snow continues, the boundary of the rock-ice interface is subjected to an ongoing shift towards higher elevations circumnavigating the 2004-2008 lava dome. It is very likely for the cave systems to follow these patterns and to continue extension towards the dome (Fig 17). This will probably happen within the next few decades, mostly controlled by
fumarolic activity. To make more precise calculations for the evolution of glaciovolcanic caves on Mount St. Helens longer time series data and additional resurveys are required. In comparison to our studies of other glaciovolcanic cave systems, the system on Mount St. Helens is likely the fastest growing of its type.

We assume that the rock-ice interface is most significant for the further evolution of cave systems and for generating circular
patterns around the dome. Most caves appear to follow the same characteristic horizontal master passage network orientation with dendritic entrance passages following the rock bed upwards to the rock-ice interface (Fig. 6). As a contact point for increased melting, due to volcanic heat or warming by sunlight, melting of lateral cave passages is expected to continue until caves have connected and a master passage has formed. The same situation probably applies to the crater rim of Mount Rainier and the East Crater Caves, the difference being that a master passage already exists. For Mount St. Helens a variety of
circumstances may be responsible that caves are not interconnected so far. First, the cave system is too juvenile to have formed a master passage connection. Second, the morphology of the lava dome floor could cause obstructions to required ventilation leading to formation of void spaces in the ice. Lastly, fumarole temperatures may be too high in some areas of the lava dome to support cave formation. Evidence is provided by Hedorah Cave and fumaroles of around 90°C which appear to stop ice accumulation in a crucial junction area that would otherwise connect three of the caves (Hedorah, Rodan, and Ghidorah).
Rodan Cave, where a laterally oriented master passage connects a section with upward trending morphology (Fig. 9), is presumably the best example that cave systems are growing and that they will certainly interconnect in one or the other way.

## 6 Conclusions

Cave surveys and climatologic investigations of glaciovolcanic caves on Mount St. Helens reveal that the systems are extremely dynamic, trending to expand in the near future. Fumarolic activity is the driving force and is responsible for the
formation and evolution of cave systems. Heat flux from fumaroles appeared to remain constant during the years of survey with minor chances being present. Although individual fumaroles arose and disappeared during our survey, concentrated areas of activity were observed to be stationary. Fumarolic activity is closely related to other features which include internal as well as external factors. The permeable crater floor with steep slopes and a complex morphology with changing elevations is most significant for the drainage of water to the hydrothermal system. Annual snowfall with maximum heights in spring and the
resulting sealing of entrances have the greatest influence on air temperatures inside the caves and induce a self-energizing process due to the accelerated melting of glacial ice. Significant seasonal cycles have been observed. Similarities but equally several differences to other glaciovolcanic systems – basically Mount Hood and Mount Rainier – could be identified. Those include differences in fumarole temperatures, morphology, the nature of the crater floor, and the hydrologic situation.
We expect that caves will interconnect during their further evolution similar to the East Crater Caves on Mount Rainier and finally circumnavigate the new lava dome. Apart from lateral extension, passages will increase with annual snow accumulation. The rock-ice interface significantly influences this process. The caves on Mount St. Helens are the most dynamic caves we have seen in the Pacific Northwest, holding hemispherical rooms and continuously changing passages. We therefore assume that a connection of systems to a master passage will probably happen within the next few decades – the absence of major

volcanic activity required. Mount St. Helens and its cave systems represents a unique natural laboratory and provides several possibilities to expand future research. As many volcanoes not only in the Cascade Volcanic Arc but worldwide host glaciovolcanic cave systems, their contribution to changes of the cryosphere from underneath and the possibilities to study a subglacial environment, investigate related hazards, or to detect evidence for renewed volcanic activity can be essential.

*Data availability*. The datasets are available from the corresponding author on request.

*Author contributions*. LS and CS conceptualized and visualized this work and led writing of the manuscript. AP supervised the work. EC took part in project administration and led expeditions. CH and TB supported fieldwork and data acquisition. CH helped with data processing and writing of the manuscript. All authors provided comments and suggested edits to the

manuscript.

*Competing interests*. The authors declare that they have no conflict of interest.

*Acknowledgements*. We thank Brent McGregor, one of the expedition leaders of Glacier Cave Explorers. We also acknowledge

cave surveyors Kathleen Graham, Scott Linn, Neil Marchington, Mark Dickey, Jessica Van Ord, and Barb Williams. We thank Jared Smith, Tom Wood, Tom Gall, Becca Stubbs, and Andrew Blackstock for safety & HAZMAT support; Aaron Messinger and Special Projects Operations for custom SCBA equipment; Lynn Moorman for enabling cave reconnaissance via Virtual Reality. Thanks to Industrial Scientific and Hilleberg for equipment support. We finally acknowledge Lee J. Florea, Glyn Williams-Jones and Thor Hansteen for advice. All authors thank the two anonymous referees for their time and effort in

reviewing the manuscript.

*Financial support*. We acknowledge support by the DFG Open Access Publication Funds of the Ruhr-Universität Bochum.




Appendix A: Mothra Cave

[Figure: Map of Mothra Cave, 2019, Crater Glacier, Mount St. Helens, Washington, USA]


**A1**: Map of Mothra Cave revealing the situation in 2019.









Appendix B: Crevasse Cave

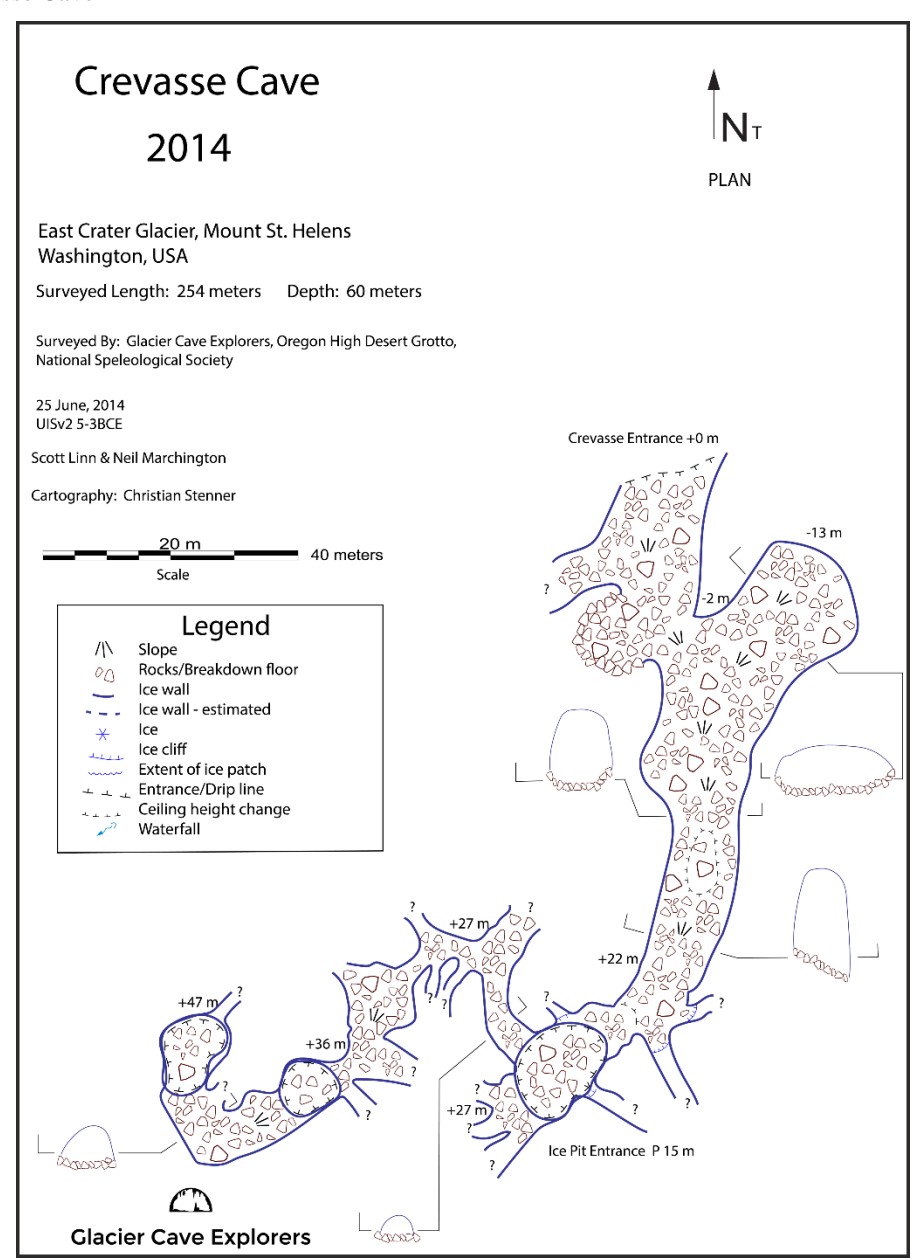


**B1**: Map of Crevasse Cave revealing the situation in 2014.




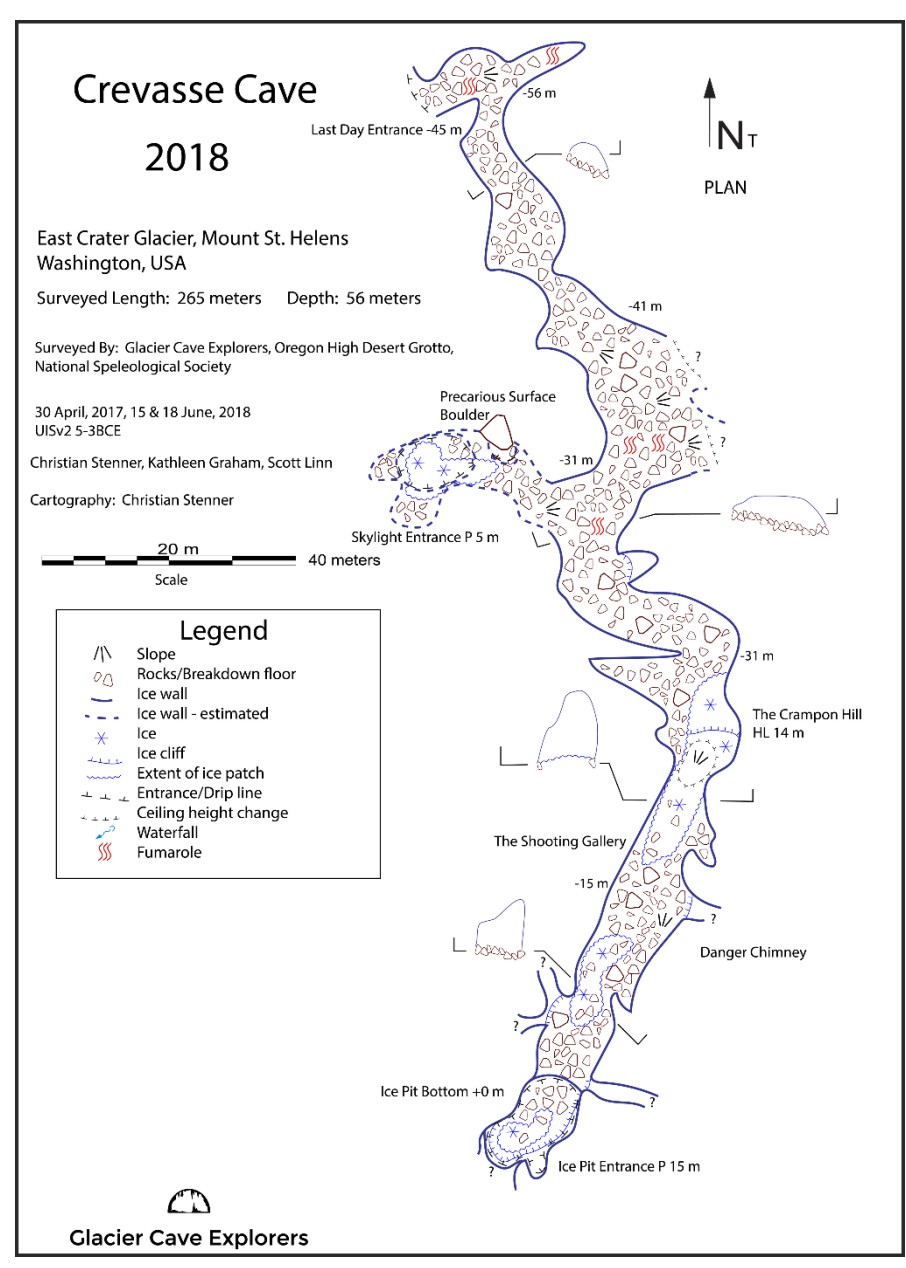

**B2**: Map of Crevasse Cave revealing the situation in 2018.






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
