# Peer review of "Formation and evolution of newly formed glaciovolcanic caves in the crater of Mount St. Helens, Washington, USA"

_The Cryosphere, 2020_

## Referee Comment (RC1) · Anonymous Referee #1 · 17 Nov 2020

* Summary

This paper introduces a cave survey in Mt. St. Helens glaciovolcanic caves (ice caves). It describes the cave geomorphologies, and reports on some temperature measurements taken therein, and some air flow data collected with smoke tracers. The authors then hypothesize about cave formation, growth, and the impact of water and where it may flow.

I'm not sure this paper is currently ready for publication in this journal. I address why under the Major Issues below. Should this work be published here or elsewhere, there are a range of minor issues that need to be fixed, of which some are highlighted below.

[Figure]

* Major

+ L68/69: "This paper provides a general summary to introduce main areas of research and only represents selected parts of our work." This sentence appears correct but I think it might be my biggest issue with the paper. I don't know what parts were unselected, but these selected parts are underwhelming. I apologize for this critique, and it may just be that I'm not the intended audience, or that I'm missing the point. But my takeaway was that this is a decent data description paper, but there isn't much else. I skimmed the Florea (in review) paper that seems closely related to this. I cannot say that this work should just be a site description or Appendix to Florea - it doesn't quite do justice to the scope of the current work. But my feeling after reading this paper a few times is that it hasn't gone very far beyond that, and so I'm not sure of the justification for a Cryosphere publication. I think it could go either way - this work could be scaled down a bit and wrapped in Florea (or in the Stenner "in prep" work), it could be published in a more appropriate journal, or it could have a significant re-write that adds new science, explains how the data presented here can be used to improve a mental, theoretical, or mathematical model of ice caves, karst systems, something cryospheric, etc. As for more appropriate journals, I thought of The Journal of Maps https://www.tandfonline.com/toc/tjom20/current. Note - after writing this I noticed the editor had similar comments upon accepting your submission.

+ L500: No. Just... no. I thought we as a community are past this. Without releasing your data this is not science, just an advertisement for some science you did. Please put all data in a data archive with DOI (I suggest Zenodo) and reference that here.

* General

+ L144: Sensors were not placed randomly but based on "interesting" areas. How did you select these? What does "climatological point of view" mean? How did you know they were interesting before you deployed the sensors? How does this selection effect your results and interpretations?

+ Table 1 and elsewhere throughout the text: Precision is not correct. Do you have high confidence that you know any value here to 1 decimal place? Length to 10 cm? Volume to 0.1 mˆ3? Anything to < 0.1 % or even 1 %?

+ L259 & Fig 10 & elsewhere: "Amplitude scaling and offset translation" - it isn't clear what methods where applied here to create Figure 10(d). Why isn't this normalized 0-1? Or -1 to 1? What is the temperature relative to (what is 0)? You should release the data that went into this figure, and the code to recreate this figure, so I can see how you implemented scaling and translation.

+ Fig 11: If you'd like to suggest things agree, then rather than show time series and ask me to squint and look for patterns, scatter these three lines against each other in a 3D or multiple 2D scatter plots.

+ L465: I suggest "main" or "primary" rather than "master"

* Minor

+ L15: "Air and fumarole" is not precise. A fumarole is an opening where air (heat, gasses, etc.) flow. Are you measuring the fumarole walls? Or the air in the cave and the air in the fumarole? How are you distinguishing between a cave entrance and a fumarole?

+ L32/33: "Import" is mentioned twice. Perhaps "usefulness" is a less hyperbolic word?

+ L35: Unclear: "abundance of loose rock with an average content of 15 %"

+ L67-68: "Our data reveal that these caves show incredible dynamic growth compared to other glaciovolcanic cave systems" I'm not sure this is shown by the current work. Certainly not "incredible". You're attempting to prove a negative and have not presented extensive survey evidence to show lack of growth elsewhere. I'm not sure what this claim adds to the paper. L460-462 are less drastic in the claim.

+ L135: ".kml" -> KML; "ArcGis" -> ArcGIS.

+ Wording and sentence flow is often complicated. Obtuse. An improved version could be easier to read. Perhaps this is just an ESL issue and I commend the first author, affiliated with a German institution, for writing English text far better than I can write any non-English language. I recommend simpler sentences in general. English co-authors should contribute more. Some examples from just one paragraph....

+ L143: What does "adjusted" mean here?

+ L145: "In an analogous manner," It isn't clear what you're analogizing here.

+ "an area with high fumarolic" <- please define fumarolic for the reader. I think this sentence might be missing some words.

+ L148: "expedition only" Only in general should come immediately before the word it is modifying. So "only left" or "only during"?

+ L150: "...a view was observed..." <- awkward phrasing.

+ L151: What is an "implementation"? Specifically the smoke tracer? Or everything described in this paragraph? Simpler phrasing: "All installations were photographed". Better: "All installations were photographed (Sobolewski, 2020)" Where Sobolewski (2020) is a data citation with DOI to the photographs that you've uploaded to Zenodo or some other data archive.

Moving on...

+ L356: "The caves contain fog and reveal a strong humidity" Not sure this is a complete sentence. Again, many sentences read rough like this.

+ L363/364: "Although we equally observed variations over time" I don't know what this means.

+ L367: "Fumaroles furthermore depict a kind of dependency" on what?

+ L408: I don't think you should cite a paper that is "In prep".

+ L429: What is the peculiarity? I couldn't find it in the following paragraph.

+ Several places: "firn ice" is used, but sometimes just "firn". I'm not sure what "firn ice" is - firn and ice are two different things. I think the correct term is just "firn".

---

## Referee Comment (RC2) · Anonymous Referee #2 · 24 Nov 2020

This paper purports to describe the formation and evolution of newly formed glaciovolcanic caves in the crater of Mt St Helens. As someone who studies both caves and glaciers, I am unable to see how this manuscript represents even a small research advance in either field and, unfortunately, it even falls short of living up to its title. The manuscript is primarily a collection of cave maps and weather data that is sandwiched between an introduction and discussion section, neither of which explain how these data sets address any specific scientific problem. Explaining how a manuscript addresses a specific research problem is probably the minimum threshold to be considered for publication in any scientific journal. This oversight is underscored by the conclusions section, which is scientifically vague and only links to a handful of caves in

the Pacific northwest. From a data analysis perspective, the authors do not go much beyond presenting weather data and cave maps. These can be important data, so I'm not being dismissive of the data, however, there is no meaningful analysis of what is driving air flow in the caves, how that airflow enlarges caves or how the ice responds to enlargement. I'm not suggesting that every paper needs a complicated numerical model in order to be accepted, but I do think that analysis needs to go beyond "Fumarole temperatures do not exceed 60.1C. This is enough heat for cave systems to form." If the paper were truly about the formation and evolution of these caves, there should, at minimum, be some back-of-the-envelope calculations of how air fluxes control cave enlargement rates. I understand and agree that glacier caves are understudied, and I'd be a bit more supportive of publication if this were the first time anyone had described glacier caves on volcanoes. I also don't want to discourage the PhD student who was the lead author on this manuscript, but Ms. Sobolewski and her academic advisers really need to start over and think critically about how their data is useful to the broader fields of speleology or glaciology before submitting to another journal. A few other more specific comments are included below: Lines 101 – Its not clear why conducting the expeditions in May and June mitigate any of these risks. It helps to be specific. The cave maps could use some work. Profile views should be included with each plan view map and the locations of all data loggers should be clearly shown in both plan and profile views (this is especially important because the authors are dealing with temperature driven air flow in some locations. As a side note, I don't see any information about outside air temperature or windspeed, both of which are going to be important in controlling air flow in the caves). Sketches should also clearly indicate different types of geologic substrate. For example, the legends indicate "breakdown" but it is not clear if the block are ice or rock. The symbol being used for ice is more often used for snow (the authors even use the same symbol for snow in their Figure 17). Also, ice isn't a particularly useful category in the context of glacier caves. Is it glacier ice or refrozen meltwater? The authors can use less space for the cave map figures if they present a single, master legend and then remove the legend and other ancillary

information from the cave map figures. Survey groups, etc. are great to include in maps used for presentations at caving clubs, etc. but they take up unnecessary space in scientific journals. Finally – when using DISTOX survey devices, why are the surveyors only making 4 cross-section measurements (L,R,U,D)? The DISTOX, especially when connected to a tablet, allows for rapidly making hundreds of splay shots which can be used to improve the sketches. . ... Finally, in foggy passages, why can't the survey team use fiberglass tape measures for making distance measurements and splay shots (old school compass and inclinometers, not mentioned in the methods, are probably fairly helpful here too). Figure 17 is interesting, and probably right, but the processes shown here are not reflected in any of the data presented in the manuscript. Lines 140-151 – Accuracy and precision of each sensor should be included, as well as the sampling rate. While smoke torches might help the team visualize flow direction while mapping, why not use anemometers for logging direction and speed? Onset makes ones that lightweight and low cost. Line 164 – Are cryospeleothems not just icicles? Line 170 – Table 1 – Presentation of volume requires a detailed explanation of how a 3D surface was put over the cave survey. I'm guessing the team just exported this information from COMPASS, but the volume derived from the cave survey described and COMPASS is very unlikely to be representative of the actual cave volume. Even within COMPASS there will be significant variability in cave volumes based on how the 3D surface is created over the survey points (COMPASS is also not the best tool for creating 3D models of cave surfaces if you use the DISTOX to create hundreds or thousands of survey points to better constrain cave shape and volume – I include this information just in case the authors want to start adopting those strategies moving forward. . ...). Lines 205 – Figure 8 – It might be more effective to show how the caves change by putting both years on the same map. . ... Also, its not clear if the rocky area on the surface is a shallow debris apron or a rocky slope. If it's the latter, is there a georeferencing error? I'm just trying to understand how the cave passages relate to the rocky substrate in the satellite image. . ...